# Sweet as Sugar—How Shared Social Identities Help Patients in Coping with Diabetes Mellitus

**DOI:** 10.3390/ijerph191710508

**Published:** 2022-08-23

**Authors:** Svenja B. Frenzel, Antonia J. Kaluza, Nina M. Junker, Rolf van Dick

**Affiliations:** 1Department of Social Psychology, Goethe University Frankfurt, 60323 Frankfurt, Germany; 2Department of Psychology, University of Oslo, 0373 Oslo, Norway

**Keywords:** social identification, diabetes mellitus, social support, stress, diabetes management, social cure

## Abstract

Social identification is health-beneficial as social groups provide social support (i.e., the social cure effect). We study this social cure effect in diabetes patients by focusing on two relevant sources of social support, namely medical practitioners (MP) and fellow patients. As both groups have diabetes-specific knowledge, we predict that sharing an identity with them provides access to specific support, which, in turn, optimizes individuals’ diabetes management and reduces diabetes-related stress. We further predict that identifying with their MP or fellow patients will be more strongly related to perceived social support among individuals with lower diabetes-specific resilience because they pay more attention to supportive cues. We tested this moderated mediation model in a two-wave study with *n* = 200 diabetes patients. Identification with the MP related to more support, which, in turn, was related to better diabetes management and less diabetes-specific stress. Identification with fellow patients related to more support; however, social support was unrelated to diabetes management and stress. Resilience only moderated the relationship between MP identification and support, as people with lower resilience levels reported more support from their MP. This study shows the importance of social identification with the MP and other diabetes patients, especially for people with lower resilience levels.

## 1. Introduction

Diabetes mellitus is a chronic disease that occurs when the beta cells located in the pancreas produce insufficient or no insulin. This shortage in insulin production is caused either by an autoimmune process that destroys the insulin-producing cells (diabetes mellitus type 1) or by occurring insulin resistance of the body cells (diabetes mellitus type 2) [1]. Insulin works like a key as it opens the cell doors and lets the glucose enter the body cells from the bloodstream. An un(der)-treated diabetes mellitus diagnosis leads to unhealthy blood glucose levels that cause serious complications, such as peripheral neuropathy (nerve damage usually affecting feet and legs), retinopathy (damage of small retinal blood vessels), and cardiovascular disease [2]. In 2021, 536.6 million people were diagnosed with diabetes mellitus worldwide with $966 billion in associated health care costs, and 6.7 million people died because of the disease or its complications [3]. However, projections paint an even more troublesome picture of the future: By 2030, the estimated diabetes prevalence will reach 643 million people worldwide, and health care costs are expected to exceed the $1 trillion mark [3,4].

Besides a possible decline in physical health, diabetes mellitus can also burden the patients’ mental health as it may induce diabetes-related distress [5]. To reduce the risk of those adverse side effects and maintain a good quality of life, patients need to manage their disease effectively and care for their health by monitoring their blood glucose and taking prescription medication or insulin [6]. In addition to adhering to the medical treatment regimen, psychosocial factors, such as social identification, can improve the well-being of patients with chronic diseases [7].

This so-called *social cure* function is captured in the *Social Identity Approach to Health* [8], which states that through identification with social groups (i.e., family, friends, fellow patients), individuals not only establish meaningful connections to other group members but also gain access to social resources, such as social support. Thereby, research suggests that social support provided by in-group members (i.e., people with whom the individual shares an identity) is especially effective because individuals evaluate such support as being well-intended (‘They help me because they sincerely care for my well-being’) [9]. Accordingly, the stronger the shared identity with fellow group members, the stronger the perception of receiving effective and benevolent social support from this group [10].

Previous studies showed that social support provided by family and friends, fellow patients, or medical staff improves diabetes management and reduces diabetes-related stress [11,12,13,14]. However, up to this point, it is unclear whether social support indeed explains the social cure phenomenon in patients with diabetes mellitus (i.e., that there is an indirect relationship between social identification and diabetes management and diabetes-related stress via social support).

With this study, we want to address this research gap. We focus on two possible types of social identification: identification with acquainted fellow patients and the primary medical practitioner. We chose these two identification types because they dispense diabetes-specific knowledge that should be particularly beneficial for individuals with diabetes mellitus. Accordingly, when individuals with diabetes mellitus identify with fellow patients, they gain access to a unique source of social support because all group members deal with similar stressors due to their diabetes diagnosis and have first-hand knowledge on how to handle these. Similarly, a shared identity with one’s medical practitioner provides access to support that is grounded in extensive knowledge about the disease. Furthermore, by focusing on identification with the medical practitioner, we test whether the identification with just one person—and not a group—can elicit the social cure effect.

Even though individuals can feel connected to fellow patients or their medical practitioner and obtain resources through them, in the end, chronic disease management is a task that is mainly performed individually and solitarily. After all, patients must come to terms with their disease by themselves. Here, research further suggests that individuals who not only accept but also gain feelings of strength and perseverance from their diagnosis are more capable of managing their disease successfully [6]. Jones et al. [15] referred to this finding as identity work. They provided evidence that a ‘survivor-identity’ (in their study, patients who perceived that their head injury has made them stronger) contributes to the recovery process. Even though diabetes mellitus is not curable but only treatable [16], these results point in the direction that accepting the disease, feeling capable of performing the required self-care behaviors, and gaining strength from it contribute to more well-being. In the present study, we refer to this as *diabetes-specific resilience*. Accordingly, the second goal of this study is to examine the interplay between social identification (with fellow patients and the medical practitioner) and people’s diabetes-specific resilience. We hypothesize that patients who report more diabetes-specific resilience are less susceptible to the social cure. Specifically, they should perceive less social support from their group memberships. Thus, we propose that diabetes-specific resilience should weaken the indirect relationship between social identification and diabetes management and diabetes-related stress, as visualized in Figure 1. We test our hypotheses in a two-wave study with *N* = 200 at Time 1.

## 2. Theoretical Background and Hypotheses

### 2.1. Social Identification with Fellow Patients

In the course of their lives, individuals feel connected to multiple social groups, such as their families, friends, work group colleagues, or supporters of a particular sports team. When feeling connected to a social group, they perceive themselves as sharing an identity with their fellow group members and pursuing similar goals, norms, and values. Significantly, they also experience stressors and difficulties as a collective matter [8]. Individuals, who live with diabetes mellitus, can sometimes feel unconnected to other healthy individuals because they have to make specific decisions (‘Should I take fewer insulin units before this walk?’) and deal with unique struggles (‘My blood sugar is low, I have to take a break!’) that others might not fully understand. Consequentially, people with diabetes mellitus indicate that they value connections with fellow patients as they share a mutual understanding and experience of what it means to have diabetes mellitus [17]. In fact, (chronically) ill patients do not have to share the same diagnosis to benefit from these social connections. The sole integration and identification with a group comprising other chronically ill individuals increased their efficacy in dealing with their chronic diseases, which improved their mental and physical health [18]. This observation is in line with the premises of the Social Identity Approach to Health because it shows that having meaningful connections (based on shared identities) with fellow patients relates to more well-being and goes beyond just being in a group of people with similar challenges.

Yet, what are the driving forces behind the link between social identity and well-being? Besides feelings of self-efficacy [18], scholars have suggested social support as a mechanism through which social identification relates to more well-being [19,20]. To be part of a social group creates a social context in which all group members are willing to provide each other with emotional, instrumental, and informational support. For example, fellow patients can share tips and recommendations about new gadgets and equipment to monitor and treat their diabetes mellitus, explain their approved strategies on how to handle glucose levels in challenging situations with one another (i.e., informational support), provide each other with dextrose or food during incidences of hypoglycemia (i.e., instrumental support), and show understanding in troublesome times (i.e., emotional support). In fact, research on peer support (i.e., individuals who are specifically trained to assist chronically ill patients) suggests that people who ‘get’ what it’s like to live with a chronic disease significantly enrich a patient’s social network by providing more appropriate and diabetes-specific support [21]. Accordingly, the more individuals identify with their fellow patients, the more they should perceive themselves to be well supported by fellow group members.

Social support, in turn, should relate to more well-being. Indeed, a meta-analysis focusing on the effectiveness of group interventions for people with diabetes mellitus type 2 backs this assumption as social support provided by fellow patients positively affected health-related self-care behaviors [14]. Regarding stress perceptions, qualitative and quantitative studies suggest that fellow patients are an essential support network that can reduce emotional distress [22,23].

Combined, these studies indicate that fellow patients can be an excellent source of social support. Accordingly, we suggest that people with diabetes mellitus who identify with fellow patients experience more social support, which in turn, should positively predict their diabetes management and reduce diabetes-related stress.

We propose that:

**Hypothesis** **1** **(H1).**
*The positive relationship between identification with fellow diabetes patients and diabetes management is mediated by social support provided by fellow diabetes patients.*


**Hypothesis** **2** **(H2).**
*The negative relationship between identification with fellow diabetes patients and stress is mediated by social support provided by fellow diabetes patients.*


### 2.2. Social Identification with the Medical Practitioner

However, not only fellow patients can provide adequate support. In fact, regularly meeting with a medical practitioner is one of several pillars contributing to a comprehensive and adequate diabetes management [24]. Research suggests that the patient–practitioner relationship should be an equal partnership that motivates patients to take an active part in their diabetes management to achieve the best treatment outcomes [25]. Jones et al., in particular, emphasized that the collaboration between patient and practitioner, also labeled a working alliance, is an essential health-contributing factor [26]. In a good working alliance, the patient and practitioner agree on the necessary tasks and treatment goals, whereas blame and judgmental behaviors are absent. These results mirror the basic premises of the Social Identity Approach to Health as they highlight the importance of the patient and their practitioner being on the same page regarding their diabetes treatment approach.

Although the majority of research on social identification has been conducted with social groups as the focus of one’s identification, initial evidence also supports our assumption that identification with a single person can also be positive [27,28]. Moreover, initial empirical evidence shows that a positive patient–practitioner relationship—as when the patient strongly identifies with their practitioner—relates to better diabetes-related self-care behaviors [29]. Those behaviors are part of an adequate diabetes management and can contribute to diabetes-related stress prevention. Accordingly, we propose that patients who identify with their medical practitioner (i.e., they perceive a meaningful connection with this person) obtain more benefits from their patient–practitioner relationship, which translates into more confidence in their diabetes management and less diabetes-related stress.

Similar to identification with fellow patients, we suggest that the underlying mechanism that explains the positive relationship between identification with the medical practitioner and well-being outcomes is social support. In particular, patients who share an identity with their medical practitioner perceive having the same values and goals with them and thus, should experience their regular meetings as more supportive. This is because they should be more open to discussing ideas and suggestions with their practitioner. Such a trusting relationship allows for sharing stress-inducing issues, struggles, and uncertainties, which the medical practitioner can take into account and respond to during the counseling sessions. In line with this reasoning, results obtained from a qualitative study show that in an equal patient–practitioner relationship, medical practitioners’ support positively affects their patients’ diabetes management [25]. Furthermore, medical practitioners’ support is generally appreciated by their patients and improves their diabetes-related self-care behaviors [12,30,31]. Regarding diabetes-related stress, Karlsen et al. [13,32] even showed that medical practitioner’s support was a better predictor of diabetes-related stress than clinical indicators, such as the average blood glucose level (HbA1c) or diabetes-related complications.

Accordingly, we propose that:

**Hypothesis** **3** **(H3).**
*The positive relationship between identification with the medical practitioner and diabetes management is mediated by social support provided by the medical practitioner.*


**Hypothesis** **4** **(H4).**
*The negative relationship between identification with the medical practitioner and diabetes-related stress is mediated by social support provided by the medical practitioner.*


### 2.3. The Social Cure Effect Is Contingent on Diabetes-Specific Resilience

We further propose that the social cure mechanisms proposed above are contingent on diabetes-related resilience. Patients who feel less in control of their health are more prone to supportive resources; for instance, they actively seek additional information from health personnel [33]. Likewise, Schulz and Schwarzer emphasized that patients who need support are more motivated to seek helping resources, which correlates with more perceived social support [34].

In light of our research context, this indicates that individuals who feel overwhelmed with their diabetes diagnosis (i.e., have a lower diabetes-specific resilience) have a higher need for assistance and, thus, should be more responsive to supportive resources from their fellow group members (i.e., fellow patients and medical practitioner) compared to patients with higher resilience levels. This is because less resilient individuals should need more social support. Thus, they find themselves in more situations in which they perceive supportive behaviors by fellow group members directed toward them. Conversely, resilient people should feel sufficiently competent to cope with occurring disease-related problems by themselves, making them less susceptible to detect positive and well-intended supportive cues in their environment (i.e., such cues for support are less relevant to them). In other words, one can only receive genuine and effective support when there is a problem to solve.

Accordingly, we propose that the individual’s diabetes-specific resilience alters the relationship between social identification and social support. Particularly, patients with a lower diabetes-specific resilience should approach their illness in an interdependent way as they rely more on the social support provided by fellow patients or their medical practitioner regarding their diabetes management and coping with diabetes-related stressors. By contrast, highly resilient patients, who incorporate their diabetes mellitus into their personal identity, approach their illness more independently when managing their diabetes and coping with stressors. Thus, they rely less on their social supportive resources.

Accordingly, we propose that:

**Hypothesis** **5** **(H5).**
*Diabetes-specific resilience moderates the indirect relationship between identification with fellow patients and diabetes management (H5a) and diabetes-related stress (H5b). Specifically, diabetes-related resilience weakens the positive relationship between identification with fellow patients and social support by fellow patients.*


**Hypothesis** **6** **(H6).**
*Diabetes-specific resilience moderates the indirect relationship between identification with the medical practitioner and diabetes management (H6a) and diabetes-related stress (H6b). Specifically, diabetes-related resilience weakens the positive relationship between identification with fellow patients and social support by fellow patients.*


## 3. Materials and Method

### 3.1. Participants and Procedure

We recruited participants via Prolific, an online platform on which researchers can share their studies and call for eligible participants. People were eligible for participation when they were of legal age and diagnosed with diabetes mellitus type 1 or 2. At the beginning of the questionnaire, we asked participants whether they knew other individuals with a diabetes diagnosis and how frequently they interact with them. We further examined whether they had a primary medical practitioner whom they would meet regularly. Data collection for Time 1 was conducted from 28–29 June 2021. We informed all participants that this study consisted of two parts and invited them to answer a second survey approximately two weeks later. Participants were incentivized with £1.25 per completed survey.

Overall, 203 people clicked the link to the first survey, and 201 completed the questionnaire. The majority of these (*n* = 189) also answered the second survey. We used three indicators to secure data quality and planned to exclude participants that were flagged on at least two of these indicators [35]: (1) average response time (flagging participants whose response times were less than 50% of the calculated median of the average response time of the sample, see Kaluza et al. [36] for a similar approach), (2) answer to an open format question (flagging those with clearly insincere answers, such as typing random letters), and (3) response patterns (flagging those who always ticked the same answer on consecutive items and scales). Based on these quality checks, no participants had to be excluded from further analyses. However, we excluded one woman because she indicated to have been diagnosed with gestational diabetes. Accordingly, the final sample consists of *N*_T1_ = 200 and *N*_T2_ = 188 participants, respectively.

The mean age was 42.18 years (*SD* = 15.24, 18–75 years) and 99 participants (49.5%) identified as women (49.5% men; 1% diverse). The majority (66.5%) were diagnosed with diabetes mellitus type 2. On average, participants had been diagnosed with diabetes mellitus for 9.92 years (*SD* = 8.31; 1–43 years; based on *n* = 199 as one participant indicated an invalid number.) Overall, 62.0% (*n* = 124) indicated to take oral medication to treat their diabetes and 46.5% (*n* = 93) to take insulin on a regular basis. At Time 1, 180 participants (90%) reported knowing at least one other person with a diabetes diagnosis. Of those, 31 participants (15.5%) indicated interacting daily with other people with diabetes mellitus; 20 (10%) specified having such interactions 2–3 days a week and 27 (13.5%) 2–4 days a month. Finally, 23 participants (11.5%) said they had only one such interaction per month, and 99 (49.5%) negated interactions with fellow patients. Moreover, 40 participants (20%) regularly engaged in a diabetes support group (online or on-site). At Time 1, 171 participants (85.5%) indicated having a main medical practitioner and reported an average of 3.13 appointments with them per year (*SD* = 3.41, 0–30; based on *n* = 199 as one person indicated an extraordinarily large value for the number of annual meetings.) We statistically compared participants who only participated at Time 1 with those who also participated at Time 2. According to *t*-test results, both samples did not differ significantly on all study-relevant variables.

### 3.2. Measures

This study was part of a larger project. In the following, we only present the relevant measures for the present paper (we further assessed depression and self-care behaviour). Unless stated otherwise, participants were asked to indicate their agreement with the given statements on a scale from 1 = *strongly disagree* to 5 = *strongly agree*. We calculated mean scores for all scales.

#### 3.2.1. Identification with Fellow Patients at Time 1

We adapted the four items by Doosje et al. [37] (sample item: ‘I am a part of this diabetes group.’) to assess identification with fellow patients at Time 1. Thereby, we instructed the participants to think about all other diabetes patients they knew and refer to these as their ‘diabetes group’. Only participants, who indicated knowing at least one person and interacting with that person at least once a month answered these items. McDonald’s Omega was 0.93.

#### 3.2.2. Identification with the Medical Practitioner at Time 1

Items to measure identification with the medical practitioner were adapted from Shamir et al. [38]. As the original purpose of the seven items was to measure identification and trust in one’s leader, we slightly changed the wording to make them fit our research context (sample item: ‘My values are similar to his/her values’). We added another item to ask participants directly how much they identify with their medical practitioner (‘I identify with him/her’). McDonald’s Omega was 0.94.

#### 3.2.3. Diabetes-Specific Resilience at Time 1

To assess diabetes-specific resilience, we adapted two items from Jones et al. [15]. Instead of referring to life after a brain injury, we asked participants whether they felt their diabetes diagnosis made them a stronger person and whether they would live a good life despite their diagnosis. McDonald’s Omega was 0.73.

#### 3.2.4. Social Support from Fellow Diabetes Patients and the Medical Practitioner at Time 2

Social support was measured with four items adapted from Haslam et al. [20]. Participants indicated whether they felt emotionally, instrumentally, and informationally supported by fellow patients (sample item: ‘Do you feel you get the help you need from this diabetes group?’) and their medical practitioner (sample item ‘Do you feel you get the emotional support you need from your practitioner?’). We collapsed all items into one overall support factor separately for fellow patients and the medical practitioner. McDonald’s Omega for social support by fellow patients was 0.95 and by the medical practitioner was 0.94.

#### 3.2.5. Diabetes Management at Time 2

We assessed participants’ diabetes management with seven items (sample item: ‘It is difficult for me to find effective solutions for problems that occur with managing my diabetes.’) obtained from the Perceived Diabetes Self-Management Scale (PDSMS) [39]. Due to an error in the questionnaire programming, we missed including the eighth item of the scale (‘I succeed in the projects I undertake to manage my diabetes.’). However, the remaining seven items should be sufficient in assessing the construct because all items are very similar, conceptualized to load on one overall factor and McDonald’s Omega was 0.90.

#### 3.2.6. Diabetes-Related Stress at Time 2

We used the Problem Areas in Diabetes Questionnaire [40] to assess diabetes-related stress (sample item: ‘Not having clear and concrete goals for your diabetes care?’). This questionnaire contains 20 statements describing potential problems associated with the diabetes treatment. Participants were asked to indicate whether they struggle with said problems on a scale ranging from 1 = *not a problem* to 5 = *serious problem*. McDonald’s Omega was 0.96.

### 3.3. Statistical Analyses

Statistical analyses were performed in SPSS v. 28 (IBM Corp., Armonk, NY, USA) [41] and Mplus v. 8.3 (Muthén & Muthén, Los Angeles, CA, USA) [42]. We specified and compared five measurement models to identify the best fitting model for our data. Based on the best fitting measurement model, we determined two structural models—the first to test the proposed mediation model and the second to test the moderated mediation model. To reduce the model complexity, we tested the hypotheses for identification with fellow patients (H1–2, H5) and the medical practitioner (H3–4, H6) separately.

First, we performed mediation analysis to test H1–H4. In each model, social identification (with fellow patients and medical practitioner, respectively) predicted social support (by fellow patients and medical practitioner, respectively). Social identification and social support then predicted diabetes-related stress and diabetes management.

Second, we specified a moderated mediation model by including diabetes-specific resilience as a moderator in the respective mediation model to test H5 and H6. Here, we created a latent interaction term by multiplying the respective form of social identification (with fellow patients and medical practitioner, respectively) with diabetes-specific resilience. We added the interaction term and diabetes-specific resilience in the analysis predicting social support. A moderated mediation would be indicated by a significant effect of the interaction term on social support (i.e., mediator). We applied the Johnson–Neyman technique to identify for which values of diabetes-specific resilience the indirect effect was significant. We standardized all variables in all analyses to simplify the interpretation and comparability of our results across all structural models. The maximum likelihood parameter estimation with robust stand errors (MLR) was used for the measurement and structural models.

### 3.4. Measurement Models

Before performing the following analyses, we excluded the item ‘Feeling unsatisfied with your diabetes physician’ from the stress scale to enhance the model fit as it was negatively correlated with items assessing identification with the medical practitioner. In Model 1, a 7-factor model, identification with fellow patients, identification with the medical practitioner, diabetes-specific resilience, social support provided by fellow patients, social support provided by the medical practitioner, diabetes management, and diabetes-related stress were all specified as first-order latent factors with all items loading on their intended a priori factor. In Model 2, a 9-factor model, all variables except the stress variables were specified as first-order latent factors with all items loading on their intended a priori factor. Stress was specified as a hierarchical second-order latent factor comprising two first-order latent factors (3 and 16 items, respectively measured first-order latent stress-factor 1 and first-order latent stress-factor 2). In Model 3, a 6-factor model, the items for measuring diabetes-related stress and diabetes management loaded on a first-order latent factor, and all other items loaded on their intended a priori factor. We specified Model 3 to test the difference between diabetes-related stress and diabetes management. Model 4, a 6-factor model, was specified to test the difference between identification with fellow patients and identification with the medical practitioner. Therefore, all identity items (i.e., identification with fellow patients and the medical practitioner) loaded on one first-order latent factor and all other variables on their intended a priori factors. Finally, Model 5 was a one-factor model with all items loading on one overall latent factor. Model fit was established with the common fit indices [43]: (1) lowest chi-square-value, (2) root mean square error of approximation (RMSEA: 0.90), (5) comparative fit index (CFI: >0.90), and (6) smaller AIC and BIC value. Significant factor loadings (*λ* > 0.40, *p* < 0.01) and small residual variances were considered as indicators for measurement quality [44]. Model fit indices are presented in Table 1. We statistically compared the models with the Satorra–Bentler scaled chi-square difference test [45]. The results showed that the intended Model 2 was superior to the other four models (see Table 1).

Model fit of Model 2 was further improved by allowing the residual variances of the following variables to covary (*χ*^2^ (1054) = 1726.61, *p* < 0.001, scaling correction factor for MLR = 1.03, RMSEA = 0.056, 90% CI: [0.052, 0.061], CFI = 0.897, TLI = 0.890, SRMR = 0.076). Identification with the medical practitioner scale: item 5 (‘He/She represents values that are important to me’) with item 6 (My values are similar to his/her values), and item 7 (‘He/she is a model for me to follow’) with item 8 (‘I identify with him/her’); diabetes management: item 3 (‘I handle myself well with respect to my diabetes’) with item 4 (‘I am able to manage things related to my diabetes as well as most other people’). Compared to Model 2, these residual correlations led to a significant improvement of the model fit (*SB-*Δ*X*^2^ = 108.73, Δ*df* = 3, Δ*c* = 1.8083, *p* < 0.001). We performed all analyses with and without the residual correlations, which did not change our results. Accordingly, we decided to report the results based on the structural models without these residual correlations.

## 4. Results

Before we conducted the main analyses, we statistically compared people with diabetes mellitus type 1 and type 2. According to *t*-test results, the samples did not differ on any study-relevant variables.

We present the descriptive statistics in Table 2 and correlations between all relevant variables in Table 3.

### 4.1. Hypotheses Testing

#### 4.1.1. Results for the Simple Mediation Analyses (Hypothesis 1–4)

Table 4 depicts the results of all structural models. In the first structural model, we tested whether social support provided by fellow patients mediated the relationship between identification with fellow patients at Time 1 and diabetes management (H1) and stress at Time 2 (H2). The model had an acceptable fit to the data (*χ*^2^ (519) = 956.94, *p* < 0.001, scaling correction factor for MLR = 1.06, RMSEA = 0.07, 90% CI: [0.060, 0.073], CFI = 0.89, TLI = 0.89, SRMR = 0.07). Our results show a non-significant indirect effect of identification with fellow patients on diabetes management via social support provided by fellow patients (*γ* = 0.10, SE = 0.06, *p* = 0.072, 95% CI [−0.009, 0.212]). Particularly, identification with fellow patients at Time 1 was indeed positively related with social support by fellow patients at Time 2 (*γ* = 0.40, SE = 0.11, *p* < 0.001, 95% CI [0.175, 0.617]). However, social support was only marginally associated with diabetes management (*γ* = 0.26, SE = 0.13, *p* = 0.051, 95% CI [−0.001, 0.513]). Therefore, H1 was not supported. Furthermore, and contradicting H2, the indirect effect from identification with fellow patients on stress via social support was not significant (*γ* = −0.03, SE = 0.04, *p* = 0.529, 95% CI [−0.111, 0.057]) as there was no association between social support by fellow patients and diabetes-related stress at Time 2 (*γ* = −0.07, SE = 0.11, *p* = 0.527, 95% CI [−0.280, 0.143]).

In the second structural model, we tested whether social support provided by the medical practitioner mediated the relationship between identification with the medical practitioner at Time 1 and diabetes management (H3) and diabetes-related stress (H4) at Time 2. The model had an acceptable fit to the data (*χ*^2^ (657) = 1296.80, *p* < 0.001, scaling correction factor for MLR = 1.08, RMSEA = 0.07, 90% CI: [0.065, 0.076], CFI = 0.88, TLI = 0.87, SRMR = 0.07). In line with H3, we found a significant indirect effect of identification with the medical practitioner on diabetes management via social support (*γ* = 0.30, SE = 0.11, *p* = 0.004, 95% CI: [0.096, 0.513]) as identification with the medical practitioner at Time 1 was related to more social support by the medical practitioner at Time 2 (*γ* = 0.69, SE = 0.07, *p* < 0.001, 95% CI: [0.546, 0.834]). Social support, in turn, was then positively associated with diabetes management at Time 2 (*γ* = 0.44, SE = 0.15, *p* = 003, 95% CI: [0.151, 0.731]). Further, and supporting H4, there was a significant indirect effect of identification with the medical practitioner on diabetes-related stress via social support (*γ* = −0.33, SE = 0.10, *p* = 0.001, 95% CI: [−0.533, −0.127]) as social support was also negatively associated with diabetes-related stress at Time 2 (*γ* = −0.48, SE = 0.15, *p* = 0.001, 95% CI: [−0.764, 0.193]).

#### 4.1.2. Results for the Moderated Mediation Analyses (Hypotheses 5–6)

To test our moderation mediation hypotheses (H5–H6), namely that diabetes-specific resilience would moderate the indirect effect of identification on diabetes management and diabetes-related stress by weakening the positive relation between identification and social support, we included diabetes-specific resilience as a moderator in the structural models.

The interaction term (identification with fellow patients x diabetes-specific resilience) at Time 1 was unrelated to social support by fellow patients at Time 2 (*γ* = 0.07, SE = 0.09, *p* = 0.472, 95% CI: [−0.114, 0.246]). This means that diabetes-specific resilience did not weaken the relationship between identification with fellow patients and social support. The index for the moderated mediation was also insignificant for diabetes management (*γ* = 0.02, SE = 0.03, *p* = 0.524, 95% CI: [−0.040, 0.078]) and stress (*γ* = −0.01, SE = 0.01, *p* = 0.627, 95% CI: [−0.029, 0.017]) as outcome variables. Accordingly, our results did not support H5a and H5b as diabetes-specific resilience did not alter the indirect effect of identification with fellow patients on diabetes management and diabetes-related stress.

Finally, we tested whether diabetes-specific resilience moderates the indirect effect of identification with the medical practitioner on diabetes management (H6a) and diabetes-related stress (H6b). Indeed, the interaction term (identification with the medical practitioner x diabetes-specific resilience) at Time 1 was negatively associated with social support at Time 2 (*γ* = −0.35, SE = 0.09, *p* < 0.001, 95% CI: [−0.532, −0.165]). This result means that the lower the diabetes-specific resilience, the stronger the positive relationship between identification and social support by the medical practitioner. In other words, individuals with higher diabetes-specific resilience seem to benefit less from a shared identity when experiencing social support.

Furthermore, the index of the moderated mediation for diabetes management as the outcome variable was significant (*γ* = −0.19, SE = 0.08, *p* = 0.023, 95% CI: [−0.346, −0.026]). The positive regression weight indicates that the indirect effect of identification on diabetes management via social support is a decreasing function of the moderator diabetes-specific resilience. Thus, the more resilience, the smaller the indirect effect. As shown in Figure 2, for resilience values = 0.85 (on a standardized scale ranging from −1 to 1), the indirect effect is just significant (*γ* = 0.15, SE = 0.07, *p* = 0.047, 95% CI: [0.002, 0.289]). However, for resilience values = 0.86 (on a standardized scale), the indirect effect is at the significance threshold (*γ* = 0.14, SE = 0.07, *p* = 0.050, 95% CI: [0.000, 0.288]), and for values > 0.86, the indirect effect is not significant anymore (*γ* = 0.14, SE = 0.07, *p* = 0.054, 95% CI: [−0.002, 0.286] for resilience values = 0.87 on a standardized scale). Accordingly, our results support H6a.

The index of moderated mediation for diabetes-related stress as outcome variable was significant (*γ* = 0.18, SE = 0.08, *p* = 0.018, 95% CI: [0.031, 0.337]). Here, the positive regression weight indicates that the negative indirect effect of identification on diabetes-related stress via social support gets smaller (i.e., closer to zero) when diabetes-specific resilience values increase. More specifically presented in Figure 3, for resilience values = 0.86 (on a standardized scale), the indirect effect is still significant (*γ* = 0.14, SE = 0.07, *p* = 0.048, 95% CI: [−0.284, −0.001]); however, for values exceeding 0.86, the indirect is no longer significant (*γ* = −0.14, SE = 0.07, *p* = 0.052, 95% CI: [−0.2843, 0.001] for resilience values = 0.87 on a standardized scale). Conclusively, our results support H6b.

## 5. General Discussion

Our primary assumption was that social support provided by fellow patients and the medical practitioner mediates the relationship between social identification (with fellow patients and the medical practitioner, respectively) and diabetes management and diabetes-related stress. Although both forms of identification were associated with more social support, only social support by the medical practitioner, in turn, positively related to diabetes management and negatively to diabetes-related stress. These mediations were stronger, the lower the individuals’ diabetes-specific resilience.

### 5.1. Theoretical Implications and Future Research

Whereas spouses and close family members are an essential source of support [46], our study shows that the support obtained from more distant and professional relationships built on a shared identity can also improve patients’ well-being. Yet, the differential findings obtained for identification with the medical practitioner and fellow patients suggests that these might fulfill specific functions and satisfy unique needs. Regular meetings with the medical practitioner have a specific medical purpose. In addition, the medical practitioner has an ‘official’ diabetes expert status—whereas fellow patients do not. This status qualifies the practitioner to advice on diabetes-related topics. Thus, from a patient’s perspective, the social support obtained from the medical practitioner might specifically and (mostly) exclusively target fundamental diabetes-related outcomes.

Conversely, acquaintances with fellow patients are casual and voluntary. Even though all interaction partners have the same chronic disease, medical and health-related topics are not necessarily on those interactions’ agenda. Even if they are discussed, they might serve a different purpose (e.g., dealing with daily hassles, such as providing practical information about technical equipment [47]). Moreover, these relationships can exist beyond the diabetes context (e.g., fellow diabetes patients are also part of the individuals’ friends groups), so that identifying with these might predict broader outcomes, such as contributing to more life satisfaction but be—as the present study’s results suggest—less predictive of diabetes-specific outcomes.

In addition to the preceding argument, patients might show different self-disclosure behaviors towards fellow patients and their medical practitioner, which could account for the missing relation between support by fellow patients and diabetes-related stress. Individuals might prefer to discuss personal and intimate struggles in a clinical setting with a professional. They might be more reluctant to open up to fellow patients because they fear being perceived as incompetent or receiving negative feedback. This explanation would mirror evidence from previous research showing that people hesitate to share relevant health information with their immediate social environment when they expect to be stigmatized, victimized, or shamed [48,49]. As a result, patients cannot receive effective support for struggles they do not share with fellow patients. However, when they decide to do so, they report it as a thoroughly positive experience and evaluate it as helpful [50]. As we only asked about the number of general acquaintances, assessing the relationship quality and the ‘patients’ wish to discuss diabetes-related topics with fellow patients in future studies seems valuable.

The different relationship functions (i.e., practitioner’s support targets fundamental diabetes-specific outcomes; fellow patients’ support is effective when dealing with daily hassles and non-diabetes related outcomes) might also account for the missing interaction effect (i.e., identification with fellow patients x diabetes-specific resilience) on social support by fellow patients. Individuals who developed a diabetes-specific resilience might be less susceptible to support that targets fundamental diabetes-related issues (i.e., support by the medical practitioner) because they already feel confident in handling their diabetes independently. Therefore, they might use meetings with their medical practitioner especially for routine checks and for receiving new prescriptions instead of discussing alternative approaches or frustrating aspects of their diabetes regimen. Nevertheless, those individuals might still acknowledge and appreciate when fellow patients can help them with technical guidance, news on diabetes-related matters, and tips and tricks for daily diabetes management. Those support acts do not interfere with their fundamental self-concept of being ‘good on their own’, which is why the association between identification with fellow patients and support by fellow patients might not have been affected by their diabetes-specific resilience. Building on this explanation, future research might want to explore in which situations fellow patients’ support is effective. Based on our results and given that fellow patients likely experience similar obstacles in their daily lives, we would expect that they are of significant help when dealing with specific diabetes-related issues and daily hassles.

On a final note, there was a main effect of diabetes-specific resilience on support by the medical practitioner (but not on support by fellow patients) that should be discussed in more detail. The indirect effect of identification with the medical practitioner on diabetes management and stress disappeared when people were highly resilient. Yet, the positive relationship between diabetes-specific resilience and support indicates that these individuals were still able to receive support from their medical practitioner—in turn benefitting their diabetes management and diabetes-related stress—yet perceiving such support was independent of their identification levels with the medical practitioner. This result implies that individuals with less diabetes-specific resilience benefit from identification with their medical practitioner to access the social resources their medical practitioner offers. By contrast, highly resilient individuals do not require such a shared identity to benefit from the support they are offered. Those people may still be able to claim the social supportive resources they need from their medical practitioner despite not defining their own identity with respect to their relationship with the medical practitioner. This interpretation mirrors an argument by Jones et al. [26] that patients who experience low vs. high barriers in their diabetes management require different care intensities from their practitioner. These results suggest that lower care intensities, such as giving instructions and educational information are only sufficient for patients with lower barriers because they feel self-efficient in implementing them in practice. In contrast, patients experiencing higher barriers might feel unequipped to do that. In fact, a similar phenomenon has been described in the cultivation hypothesis, which states that self-efficient individuals cultivate (i.e., maintain and enhance) their social resources by initiating social exchange [51]. In other words, their ‘hands-on’ mentality enables them to mobilize their social resources when they need support [52].

Overall, the pattern of results suggests that the social cure effect might depend on individual characteristics—at least for some types of social identification. Building on these findings, we encourage researchers to test whether similar patterns are observed in other contexts (such as the work context) and dyadic relationships (such as the patient–partner relationship) and which other individual factors might be relevant for obtaining the social cure effect. For instance, according to attachment theory, individuals form attachment styles based on their early experiences with caregivers. People with a secure attachment are comfortable with being cared for by others, whereas individuals with a dismissing attachment style are more self-reliant and perceive interdependence as uncomfortable. Therefore, we would expect that patients with a secure attachment style are more susceptible to the social cure because they appreciate receiving social support [53].

### 5.2. Practical Implications

The present research also has important practical implications. In particular, based on the finding that sharing a social identity with the medical practitioner is of particular importance for diabetes management and diabetes-related stress, we would advise people with diabetes mellitus to look for a primary medical practitioner with whom they share goals, values, and norms regarding treatment strategies, even when it requires changing their current practitioner. This advice is of particular importance for patients who feel overwhelmed with their diagnosis and the necessary treatment (i.e., those who have lower diabetes-specific resilience values) because they especially benefit from ‘being on the same page’ as their medical practitioner. Likewise, we encourage practitioners to develop a shared social identity with their patients. One particular helpful way to do so would be to emphasize the similarities they share with their patient [54].

### 5.3. Limitations

Our study is, of course, not without limitations. First of all, our results do not imply causal relationships. This means that we cannot rule out whether the proposed mechanism (i.e., mediation via social support) also works the other way around. For instance, patients suffering from high diabetes-related stress might feel so overwhelmed and hopeless that they cannot accept any form of support and withdraw from any social relationships associated with their diabetes diagnosis. Thus, they would avoid contact with fellow patients and their medical practitioner. Alternatively, highly stressed individuals might be particularly focused on their medical practitioner because they require intense supervision. Indicative of this assumption is the unexpected positive direct effect of identification with the medical practitioner on diabetes-related stress. After controlling for social support by the medical practitioner, a shared identity was, in fact, stress inducing. Accordingly, it seems meaningful that future studies test the proposed moderated serial mediation with a longitudinal study design and an appropriate sample size. This allows for testing the variables in the proclaimed order and to control for baseline measures. Second, we only used self-measures to assess all study variables and strongly recommend that future studies include objective and clinical measures, such as average blood glucose levels (i.e., HbA1c value) as dependent variables. By doing so, future research could rule out the common-method bias, which describes the phenomenon that response variance and associations between variables result from the (self-report) instruments used [55]. To mitigate this bias, we used a competing measurement model approach that confirmed the statistical independence of our study variables and partially measured the predictor and the criterion variables at two different time points. Third, the reported model fits of our structural models are acceptable but not excellent. However, we statistically compared five measurement models and used the best fitting model for specifying our structural models. Finally, we did not assess whether participants are generally satisfied with the support they receive, even though higher support satisfaction is—naturally—a better shield against the adverse effects of diabetes-related burden than lower satisfaction [56]. Along those lines, we also know from previous work that people differ in their preferences regarding the intensity of the support they receive [57], which is something we did not control.

## 6. Conclusions

This study identified social support by the medical practitioner as the underlying mechanism behind social identification and well-being parameters in patients with diabetes mellitus. Most importantly, our study expands previous research on the social identity–well-being link because it shows that the identification with a single person, here the medical practitioner, is sufficient in eliciting the social cure effect. Finally, we were able to show that diabetes-specific resilience establishes boundary conditions for the social cure effect. Notably, highly resilient individuals are less susceptible to the social cure regarding their diabetes management and coping with diabetes-related stress.

## Figures and Tables

**Figure 1 ijerph-19-10508-f001:**
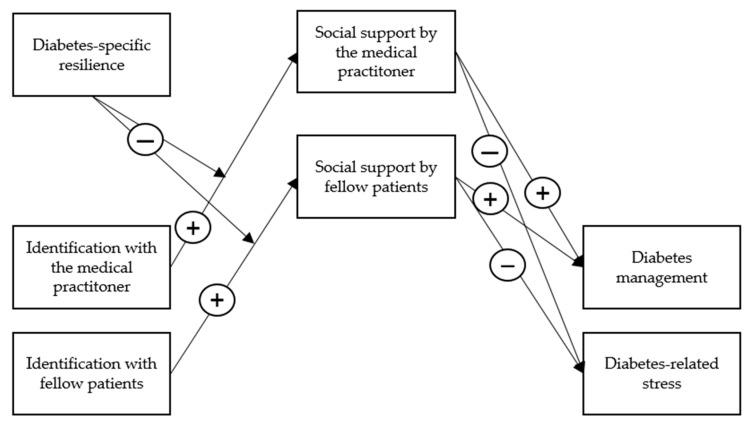
Overall research model. Identification with fellow patients and the medical practitioner and diabetes-specific resilience are measured at Time 1; all other variables are measured at Time 2 (4 weeks later).

**Figure 2 ijerph-19-10508-f002:**
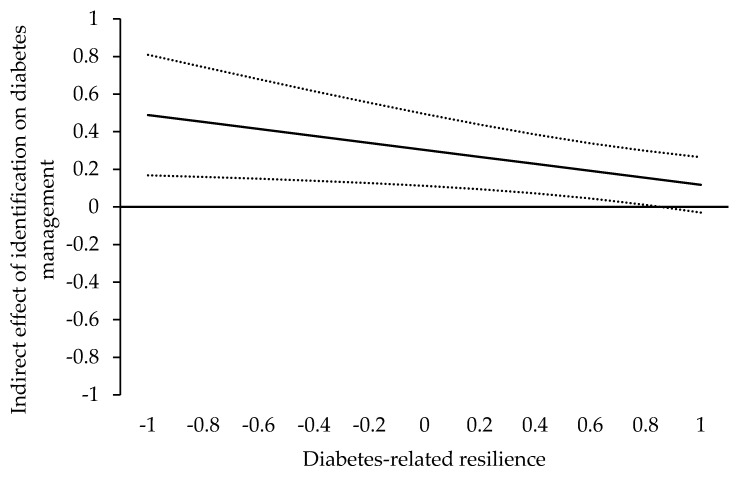
Visual presentation of the Johnson–Neyman Output for the indirect effect of identification with the medical practitioner on diabetes management. The *y*-axis depicts the indirect effect; the *x*-axis depicts the values for diabetes-specific resilience. The solid line represents the strength of the indirect effect; the dotted lines represent the lower and upper levels of the 95% confidence interval.

**Figure 3 ijerph-19-10508-f003:**
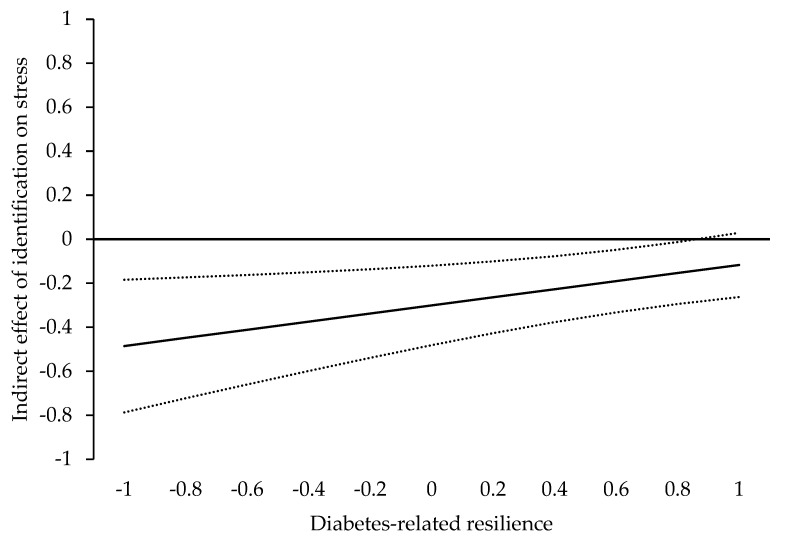
Visual presentation of the Johnson–Neyman Output for the indirect effect of identification with the medical practitioner on diabetes-related stress. The *y*-axis depicts the indirect effect; the *x*-axis depicts the values for diabetes-specific resilience. The solid line represents the strength of indirect effect; the dotted lines represent the lower and upper levels of the 95% confidence interval.

**Table 1 ijerph-19-10508-t001:** Competing measurement models.

Model	*χ^2^*	*df*	*c*	CFI	TLI	RMSEA	SRMR	RMSEA 95% CI		AIC	BIC	Model Comparison	*SB-*Δ*X*^2^	Δ*df*	Δ*c*
								LL	UL						
Model 1	1987.86 *	1059	1.0348	0.858	0.848	0.066	0.077	0.062	0.071	19,477.79	20,021.71				
Model 2	1912.84 *	1057	1.0354	0.869	0.860	0.064	0.076	0.059	0.068	19,404.94	19,955.76	Model 2 vs. Model 1	106.57 *	2	0.7177
Model 3	2274.68 *	1065	1.0396	0.815	0.804	0.075	0.088	0.071	0.080	19,773.20	20,297.63	Model 2 vs. Model 3	240.95 *	8	1.5945
Model 4	2313.39 *	1065	1.0342	0.809	0.797	0.077	0.098	0.072	0.081	19,800.85	20,325.29	Model 2 vs. Model 4	470.45 *	8	0.8756
Model 5	4786.44 *	1080	1.0418	0.432	0.407	0.131	0.193	0.127	0.135	22,364.64	22,839.59	Model 2 vs. Model 5	2250.10 *	23	1.3359

*Note.* * *p* < 0.001.

**Table 2 ijerph-19-10508-t002:** Descriptive statistics of all study-relevant variables.

Variables	*N*	*M*	*SD*	*SK*	*RKU*
Identification with fellow patients (Time 1)	101	3.49	1.02	−0.34	−0.14
Identification with the medical practitioner (Time 1)	171	3.67	0.90	−0.56	0.01
Support from fellow patients (Time 2)	112	3.13	1.06	−0.11	−0.49
Support from the medical practitioner (Time 2)	162	3.82	1.03	−0.94	0.37
Diabetes-related stress (Time 2)	188	2.40	0.90	0.42	−0.71
Diabetes management (Time 2)	188	3.47	0.90	−0.12	−0.68
Diabetes-specific resilience (Time 1)	200	3.22	1.05	−0.16	−0.71

*Note. N* (number of participants), *M* (mean), *SD* (standard deviation), *SK* (skewness), *RKU* (kurtosis).

**Table 3 ijerph-19-10508-t003:** Correlations between all study-relevant variables.

	1	2	3	4	5	6	7	8	9	10	11	12	13	14
1 Age														
2 Diabetes type §	−0.47 ***													
3 Years since diagnoses	0.13	0.41 ***												
4 Number of fellow patients as social contacts	−0.14	0.15 *	0.11											
5 Contact frequency with fellow patients ¶	0.10	0.03	−0.04	−0.11										
6 Visiting diabetes support group ǁ	−0.28 ***	0.33 ***	0.10	0.10	−0.14									
7 Treatment by medical practitioner ǁ	0.13	−0.01	0.02	0.10	−0.11	0.14								
8 Amount of annual meetings with MP	−0.21 **	0.20 **	−0.13	0.03	−0.22 **	0.10	0.05							
9 Identification with fellow patients (Time 1)	0.04	0.05	0.01	0.01	−0.29 **	0.15	0.18	0.18						
10 Identification with the medical practitioner (Time 1)	−0.02	0.05	−0.10	0.10	−0.12	−0.03	0.05	0.24 ***	0.14					
11 Support from fellow patients (Time 2)	−0.07	0.06	0.01	0.04	−0.36 ***	0.13	0.21 *	0.24 *	0.39 ***	0.31 **				
12 Support from the medical practitioner (Time 2)	0.05	−0.01	−0.13	0.09	−0.08	−0.11	0.10	0.24 **	0.14	0.73 ***	0.30 **			
13 Diabetes-related stress (Time 2)	−0.28 **	0.13	−0.14	−0.04	0.02	0.17 *	−0.02	0.13	−0.07	−0.09	−0.04	−0.24 **		
14 Diabetes management (Time 2)	0.18 *	0.00	0.12	0.12	−0.02	−0.08	0.05	0.06	0.06	0.26 ***	0.22 *	0.33 ***	−0.69 ***	
15 Diabetes-specific resilience (Time 1)	−0.07	0.10	0.02	0.13	−0.15 *	0.13	0.10	0.27 **	0.27 **	0.37 ***	0.31 ***	0.36 ***	−0.37 ***	0.52 ***

*Note.* § 0 = type 2, 1 = type 1; ¶ higher values indicate fewer interactions; ǁ 0 = no, 1 = yes; * *p* < 0.05; ** *p* < 0.01; *** *p* < 0.001.

**Table 4 ijerph-19-10508-t004:** Results of the structural equation models.

	Fellow Patients			Medical Practitioner		
	Social Support by Fellow Patients	Diabetes Management	Diabetes-Related Stress	Social Support by the Medical Practitioner	Diabetes Management	Diabetes-Related Stress
Simple Mediation Model						
Social identification	0.40 (0.11) ***	−0.06 (0.12)	−0.04 (0.11)	0.69 (0.07) ***	−0.06 (0.13)	0.20 (0.11)
Social support	--	0.26 (0.13)	−0.07 (0.11)	--	0.44 (0.14) **	−0.48 (0.15) **
Moderated Mediation Model						
Social identification	0.32 (0.12) **	−0.01 (0.15)	−0.09 (0.13)	0.57 (0.08) ***	−0.14 (0.15)	0.24 (0.12) *
Social support	--	0.29 (0.15) *	−0.09 (0.12)	--	0.53 (0.16) **	−0.53 (0.15) **
Diabetes-related resilience	0.25 (0.18)	--	--	0.35 (0.12) **	--	--
Interaction term	0.07 (0.09)	--	--	−0.35 (0.09) ***	--	--

*Note.* * *p* < 0.05, ** *p* < 0.01, *** *p* < 0.001.

## Data Availability

The data presented and analyzed in this study can be found here: https://osf.io/4a58p/?view_only=de47891e957b4abd8dc49393d68c1f24.

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
