# Peer review of "Sweet as Sugar—How Shared Social Identities Help Patients in Coping with Diabetes Mellitus"

_ijerph, 2022, doi:10.3390/ijerph191710508_

Round 1
Reviewer 1 Report
Page 1: Should be “…enter the bloodstream.”
Page 1: Should be “…as it may induce diabetes-related distress”
Page 2: Should be “because they dispense diabetes-specific knowledge…”
Page 3: Should be “…they perceive themselves as sharing an identity with…”
Page 3: Would be better to say “This observation is in line with the premises of the Social Identity Approach To Health because it
Page 4: Should be “…the more they should perceive themselves to be well supported by fellow
group members.”
Page 4: Should be: “which in turn should positively predict their diabetes management”
Page 4 and 5: ‘by’ would be better than ‘via’ in Hypothesis 1, Hypothesis 2, Hypothesis 3, and Hypothesis 4.
Page 4: Should be “In fact, regular meetings with a medical practitioner is one of several pillars…”
Page 4: Should be “as they highlight the importance of patient and practitioner being on the same page…”
Page 4: It would be good to cite some evidence of previous work that has examined strength of identification with only one person-is this a genuine group membership, or is it just an interpersonal relationship? Perhaps drawing on the therapeutic alliance literature here would help. I question whether conceptualising the doctor-patient relationship as leader and follower is helpful or appropriate: the importance of equality in healthcare relationships and therapeutic alliances is often highlighted in the literature. Indeed, the authors themselves later highlight the importance of equality in this relationship (“In line with this reasoning, results obtained from a qualitative study show that in an equal patient‐practitioner relationship, medical practitionersʹ support positively affects their patientsʹ diabetes management”).
Page 6: Should be “t‐Test results showed that the samples did not differ significantly on any study‐ relevant variables.”
Page 6: The example Identification with Fellow Patients item suggests that this measure is actually assessing strength of identification with a diabetes support group, rather than identification with all other diabetes patients. It would be useful for the authors to clarify this, and to state exactly what construct this scale is intended to measure.
Page 7: Did the authors measure Social Support, Diabetes Management, and Diabetes-Related Stress at T1? For longitudinal analysis, the T1 versions of the T2 variables need to be controlled for in the analysis. If the authors have not measured these mediator and outcome variables at T1 then I think it would be appropriate to mention this as a limitation in the Discussion.
Page 14: Should be “’hands-on’ mentality”
Page 14: What does “Accordingly, they do not have the impression to fight on two fronts.” mean?
Reviewer 2 Report
Method- Participants & Procedure
- What is the meaning of Prolific? Please, explain it briefly, and explain how you recruited participants via Prolific.
- The third paragraph (except “We statistically compared participants who only participated at Time 1 with those who also participated at Time 2. T‐Test results show that both samples did not differ significantly on all study‐relevant variables”). This section is about results and must be moved to result section.
-It is unclear where and when this study was conducted?
- The year of study must be added.
- Characteristics of participants must be added.
- Please add practical implications about the finding of resilience.
- please suggest future studies.
Reviewer 3 Report
I really liked the internal coherence and the way the hypotheses (theoretical model of analysis) were built and tested. The text is very well grounded, and the authors show experience in the investigation of mental representations associated with the disease diabetes. The text is a plea for familiarizing patients with the terms and characteristics of diabetes in order to increase their ability to cope with and solve the issues and problems they face in their daily lives. In this analysis, two types of patients are considered according to the type of diabetes they have, and it is assumed that the type of diabetes is reflected in the mental representations patients have about the disease and about themselves, leading to an attitude of more or less sharing either with other patients or with the official disease care services.
The hypothesis test shows exactly that each case and each person is different, according to characteristics of resilience, independence, social help-seeking, help-seeking only with health professionals, so that the construction of a social identity linked to diabetes is not absolutely necessary in all cases to access (take initiative/seek) supportive resources: "Conclusively, this means that a shared identity is helpful - but not in all cases necessary - to access supportive resources. In the case of patients of diabetes-mellitus the identification with the medical practioner is sufficient to ensure "social cure effect". The importance of the presence and connection between patient and medical practitioner is thus emphasized.
The demonstration of the greater or lesser (comparative) adequacy between 5 structural equation models provides solid statistics, which hopefully will not be boring for the readers. In any case, this study can be a model for other authors seeking similar methodological analyses.

Round 2
Reviewer 1 Report
I feel that the authors have addresses my comments thoughtfully and thoroughly, and that the manuscript stronger as a result.